# TOKEN POOLING IN VISION TRANSFORMERS

## ABSTRACT

Despite the recent success in many applications, the high computational requirements of vision transformers limit their use in resource-constrained settings. While many existing methods improve the quadratic complexity of attention, in most vision transformers, self-attention is not the major computation bottleneck, *e.g.*, more than 80% of the computation is spent on fully-connected layers. To improve the computational complexity of *all* layers, we propose a novel token downsampling method, called *Token Pooling*, efficiently exploiting redundancies in the images and intermediate token representations. We show that, under mild assumptions, softmax-attention acts as a high-dimensional low-pass (smoothing) filter. Thus, its output contains redundancy that can be pruned to achieve a better trade-off between the computational cost and accuracy. Our new technique accurately approximates a set of tokens by minimizing the reconstruction error caused by downsampling. We solve this optimization problem via cost-efficient clustering. We rigorously analyze and compare to prior downsampling methods. Our experiments show that Token Pooling significantly improves the cost-accuracy trade-off over the state-of-the-art downsampling. Token Pooling is simple and effective and can benefit many architectures with global attention. Applied to DeiT, it achieves the same ImageNet top-1 accuracy using 42% fewer computations.

**NEW**

## 1 INTRODUCTION

Vision transformers (Dosovitskiy et al., 2020; Touvron et al., 2021; Liu et al., 2021; Heo et al., 2021; Zheng et al., 2021) have demonstrated state-of-the-art results in many vision applications, from image classification to segmentation. However, the high computational cost limits their use in resource-restricted, real-time, or low-powered applications. While most prior work in Natural Language Processing (NLP) improve the time-complexity of attention (Tay et al., 2020b; Ilharco et al., 2020), in vision transformers the main computation bottleneck is the fully-connected layers, as we show in §3.1. The computational complexity of these layers is determined by the number of tokens and their feature dimensionality. While reducing the dimensionality improves computational cost, it sacrifices model capacity and often significantly deteriorates the accuracy of the model. On the other hand, since images often contain mostly smooth surfaces with sparsely located edges and corners, they contain similar (and thus redundant) features. Moreover, we show that, under mild assumptions, softmax-attention is equivalent to low-pass filtering of tokens and thereby produces tokens with similar features, as empirically observed by Goyal et al. (2020) and Rao et al. (2021). This redundancy in representations suggests that we can reduce the number of tokens, *i.e.*, downsampling, without a significant impact to the accuracy, achieving a better cost-accuracy trade-off than reducing feature dimensionality alone.

Downsampling is widely used in Convolutional Neural Network (CNN) architectures to improve computational efficiency, among other purposes. Given a grid of pixels or features, downsampling gradually reduces the grid dimensions via combining neighboring vertices on the grid. The prevailing max/average pooling and sub-sampling are examples of (spatially uniform) *grid-downsampling* that only uses locations on the grid to decide which vertices to combine. Such methods do not efficiently address non-uniformly distributed redundancy in images and features (Recasens et al., 2018; Marin et al., 2019). Unlike CNNs that require grid preservation, transformers allow a wider range of nonuniform data-aware downsampling layers, where a better operator can be designed.

We propose *Token Pooling*, a novel nonuniform data-aware downsampling operator for transformers efficiently exploiting redundancy in features. See the illustration and performance metric in Fig-

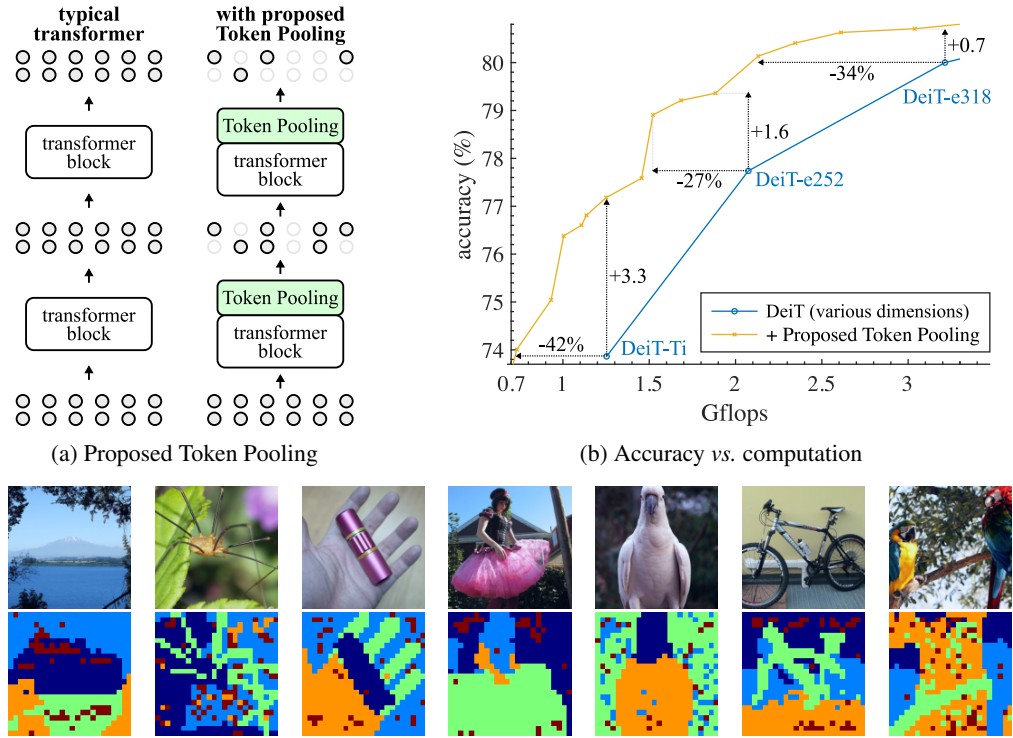

(a) Proposed Token Pooling

(b) Accuracy *vs.* computation

(c) Token Pooling via cluster analysis of token representations.

Figure 1: (a) We propose Token Pooling, a novel token downsampling method, for visual transformers. (b) The proposed method achieves a state-of-the-art trade-off between accuracy and computation. (c) Token Pooling uses cluster analysis to aggregate information from individual tokens automatically. We show the input images and the token clusters at the 6-th layer of DeiT-S.

ures 1a & 1b. Motivated by nonuniform sampling and image compression (Marvasti, 2012; Unat et al., 2009; Belfor et al., 1994; Rabbani, 2002), we formulate token downsampling as an optimization problem that minimizes the reconstruction error caused by downsampling. We show that clustering algorithms, K-Means and K-Medoids, efficiently solve this problem, see illustration in Figure 1c. To the best of our knowledge, we are the first to use this formulation and simple clustering analysis for token donwsampling in transformers. We compare with various prior downsampling techniques, including grid-downsampling (Pan et al., 2021; Liu et al., 2021; Wang et al., 2021) and token pruning (Goyal et al., 2020; Rao et al., 2021). The proposed Token Pooling outperforms existing methods and provides the best trade-off between computational cost and classification accuracy.

**Contributions.** The paper makes the following contributions:

- We conduct an extensive study of prior downsampling techniques for visual transformers by comparing their computation-accuracy trade-offs.

- We analyze the computational cost of vision-transformer components and the limitations of the prior score-based downsampling methods. We also show that attention layers behave like low-pass filtering and thus produce redundant tokens.

- Motivated by the redundancy in images and features, we propose a novel token downsampling technique, Token Pooling, for transformers with global attention via error minimization and **NEW** achieve a significant improvement in the computation-accuracy trade-off.

## 2 RELATED WORK

In this section, we introduce vision transformers, and review existing methods that improve the efficiency of transformers including existing token-downsampling methods.

## 2.1 VISION TRANSFORMERS

Vision transformers (Dosovitskiy et al., 2020; Touvron et al., 2021; Heo et al., 2021; Liu et al., 2021; Pan et al., 2021) utilize the transformer architecture that is originally designed for NLP by Vaswani et al. (2017) and further popularized by Radford et al. (2018) and Devlin et al. (2019). In a high level, a vision transformer is a composition of $L$ transformer blocks that take a set of input tokens and return another set of output tokens. In vision, input tokens are features representing individual non-overlapping image patches. To perform classification, a classification token is inserted to estimate the probabilities of individual classes. To achieve the state-of-the-art, ViT (Dosovitskiy et al., 2020) used pretraining on JFT-300M, a proprietary dataset much larger than standard ImageNet1k (Deng et al., 2009). Recently, DeiT (Touvron et al., 2021) achieved state-of-the-art results with advanced training on ImageNet1k only.

Let the set of tokens at depth $l$ be $\mathcal{F}^l = \{\boldsymbol{f}_0^l, \ldots, \boldsymbol{f}_N^l\}$ where $\boldsymbol{f}_i^l \in \mathbb{R}^M$ is the feature values of the $i$-th token. A typical transformer block $\phi$ at depth $l$ processes $\mathcal{F}^l$ by a Multi-head Self-Attention (MSA) layer and a point-wise Multi-Layer Perceptron (MLP). Let matrix $\boldsymbol{F} \in \mathbb{R}^{N \times M}$ be a row-wise concatenation of tokens $\mathcal{F}^l$. Then[1],

$$\phi(\boldsymbol{F}) = \mathrm{MLP}(\mathrm{MSA}(\boldsymbol{F})) \qquad \text{such that} \qquad (1)$$

$$\mathrm{MSA}(\boldsymbol{F}) = [\boldsymbol{O}_1, \boldsymbol{O}_2, \ldots, \boldsymbol{O}_H]\boldsymbol{W}^O \qquad (2)$$

where $H$ is the number of heads, matrix $\boldsymbol{W}^O \in \mathbb{R}^{M \times M}$ is a learnable parameter of the block, $[,]$ is column-wise concatenation and $\boldsymbol{O}_h \in \mathbb{R}^{N \times d}$ is the output of $h$-th attention head for $d = M/H$:

$$\boldsymbol{O}_h = \boldsymbol{A}_h \boldsymbol{V}_h \quad \text{such that} \quad \boldsymbol{A}_h = \mathrm{softmax}(\boldsymbol{Q}_h \boldsymbol{K}_h^\top / \sqrt{d}) \in \mathbb{R}^{N \times N}. \qquad (3)$$

Keys $\boldsymbol{K}_h$, queries $\boldsymbol{Q}_h$ and values $\boldsymbol{V}_h$ are linear projections of the input tokens (QKV projections):

$$\boldsymbol{Q}_h = \boldsymbol{F}\boldsymbol{W}_h^Q, \quad \boldsymbol{K}_h = \boldsymbol{F}\boldsymbol{W}_h^K, \quad \boldsymbol{V}_h = \boldsymbol{F}\boldsymbol{W}_h^V \qquad (4)$$

where $\boldsymbol{W}_h^Q \in \mathbb{R}^{M \times d}$, $\boldsymbol{W}_h^K \in \mathbb{R}^{M \times d}$, $\boldsymbol{W}_h^V \in \mathbb{R}^{M \times d}$ are learnable linear transformations. Note, the number of tokens is not affected by the transformer blocks, *i.e.*, $|\mathcal{F}^{l+1}| = |\mathcal{F}^l|$.

## 2.2 EFFICIENT TRANSFORMERS

Similar to many machine learning models, the efficiency of transformers can be improved via meta-parameter search (Howard et al., 2017; Tan & Le, 2019), automated neural architecture search (Elsken et al., 2019; Tan et al., 2019; Wu et al., 2019), manipulating the input size and resolution of feature maps (Paszke et al., 2016; Howard et al., 2017; Zhao et al., 2018), pruning (LeCun et al., 1990), quantization (Jacob et al., 2018), and sparsification (Gale et al., 2019), *etc*. For example, Dosovitskiy et al. (2020) and Touvron et al. (2021) obtain a family of ViT and DeiT models, respectively, by varying the input resolution, the number of heads $H$, and the feature dimensionality $M$. Each of the models operates with a different computational requirement and accuracy. In the following, we review techniques developed for transformers.

### 2.2.1 EFFICIENT SELF-ATTENTION

The softmax-attention layer (3) has a quadratic time complexity w.r.t. the number of tokens, *i.e.*, $\mathcal{O}(N^2)$. In many NLP applications where every token represents a word or a character, $N$ can be large, making attention a computation bottleneck (Dai et al., 2019; Rae et al., 2019). While many works improve the time complexity of attention layers, as we will see in §3.1, they are not the bottleneck in most current vision transformers.

The time complexity of an attention layer can be reduced by restricting the attention field-of-view and thus imposing sparsity on $\boldsymbol{A}_h$ in (3). This can be achieved using the spatial relationship between tokens in the image/text domain (Parmar et al., 2018; Ramachandran et al., 2019; Qiu et al., 2020; Beltagy et al., 2020; Child et al., 2019; Zaheer et al., 2020) or based on token values using locality-sensitive hashing, sorting, compression, *etc*. (Kitaev et al., 2020; Vyas et al., 2020; Tay et al., 2020a; Liu et al., 2018; Wang et al., 2020; Tay et al., 2021). Prior works have also proposed attention mechanisms with lower time complexity, *e.g.*, $\mathcal{O}(N)$ or $\mathcal{O}(N \log N)$ (Katharopoulos et al., 2020;

Peng et al., 2021; Choromanski et al., 2021; Tay et al., 2021). Roy et al. (2021) cluster queries and **NEW** keys to sparsify attention matrices to speed-up the attention, but they do not downsample the tokens.

Note that the goal of these methods is to reduce the time complexity of the attention layer—the number of tokens remains the same across the transformer blocks. In contrast, our method reduces the number of tokens *after* attention has been computed. Thereby, we can utilize these methods to further improve the overall efficiency of transformers.

Recently, Wu et al. (2020) proposed a new attention-based layer that learns a small number of query vectors to extract information from the input feature map. Similarly, Wu et al. (2021) replace self-attention with a new recurrent layer that outputs a smaller number of tokens. In comparison, our method directly minimizes the token reconstruction error due to token downsampling. Also, our layer has no learnable parameters and can be easily incorporated into existing vision transformers.

### 2.2.2 DOWNSAMPLING METHODS FOR TRANSFORMERS

**Grid-downsampling.** The input tokens of the first vision transformer block are computed from image patches. Therefore, even though transformers treat tokens as a set, we can still associate a grid to the tokens using their initial locations on the image. The regular grid structure allows typical downsampling methods, such as max/mean pooling, uniform sub-sampling. For example, Liu et al. (2021), Heo et al. (2021), and Wang et al. (2021) use convolutions with stride to downsample the feature maps formed by the tokens.

**Score-based token downsampling.** In the area of NLP, Goyal et al. (2020) introduce the idea of dropping tokens. Their approach, called PoWER-BERT, is based on a measure of *significance score*, which is defined as the total attention given to a token from all other tokens. Specifically, the significance scores of all tokens in the $l$-th transformer block, $s^l \in \mathbb{R}^N$, is computed by

$$s^l = \sum_{h=1}^{H} {A_h^l}^\top \mathbf{1}, \tag{5}$$

where $A_h^l$ is the attention weights of head $h$ defined in (3). They only pass $K_l$ tokens with the highest scores in $s^l$ to the next transformer block. The pruning is performed on all blocks.

PoWER-BERT is trained using a three-stage process. First, given a base architecture, they pretrain a model without pruning. In the second stage, a soft-selection layer is inserted after each transformer block, and the model is finetuned for a small number of epochs. Once learned, the number of tokens to keep, $K_l$, for each layer is computed from the soft-selection layers. Last, the model is finetuned again with the tokens pruned using the $K_l$ from the second stage. See Goyal et al. (2020) for details.

Recently, Rao et al. (2021) proposed Dynamic-ViT that also uses scores to prune tokens. Unlike PoWER-BERT, which computes significance scores from attention weights, Rao et al. use a dedicated sub-network with learned parameters. The method requires knowledge distillation, Gumbel-Softmax, and straight-through estimators on top of the DeiT training.

We will analyze the limitations of score-based methods in §3.2. Note that while downsampling **NEW** tokens can complicate some downstream tasks, *e.g.*, semantic segmentation and pose estimation, many techniques Wu et al. (2020); Marin et al. (2019) have been proposed to deal with the problem.

## 3 ANALYSIS

This section addresses three questions. First, it identifies the computational bottleneck of vision transformers. Second, it discusses the limitations of score-based downsampling. Third, it analyzes how the softmax-attention affects the redundancy in tokens.

### 3.1 COMPUTATION ANALYSIS OF VISION TRANSFORMERS

We analyze the time complexity and computational costs (measured in flops) of commonly used vision transformers, namely ViT and DeiT. We breakdown the computation into four categories:

---

[1]For compactness, we omit layer norm and skip-connections, see Dosovitskiy et al. (2020) for details.

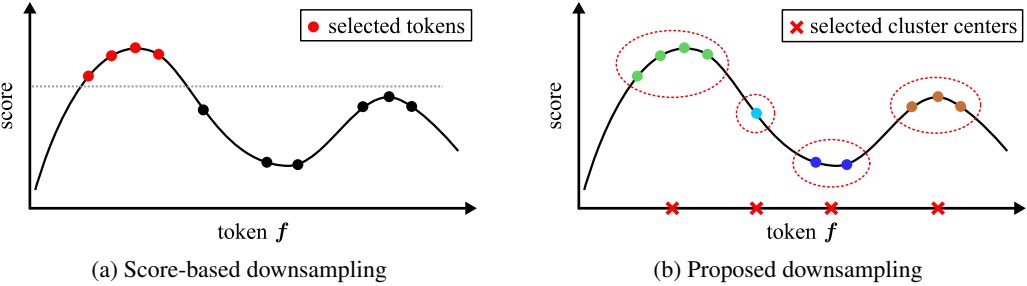

| Layer | Complexity | Computation ($10^9$ Flops) | | | |
|---|---|---|---|---|---|
| | | ViT-B/384 ($N=577$) | ViT-B ($N=197$) | DeiT-S ($N=197$) | DeiT-Ti ($N=197$) |
| softmax-attention | $\mathcal{O}(LN^2M)$ | 6.18 | 0.72 | 0.36 | 0.18 |
| QKV projections | $\mathcal{O}(LNM^2)$ | 12.25 | 4.18 | 1.05 | 0.26 |
| O projection | $\mathcal{O}(LNM^2)$ | 4.08 | 1.39 | 0.35 | 0.09 |
| Multi-linear Perceptron | $\mathcal{O}(LNM^2)$ | 32.67 | 11.15 | 2.79 | 0.70 |
| Total | $\mathcal{O}(LNM(M+N))$ | 55.5 | 17.6 | 4.6 | 1.3 |

Table 1: Time complexity and computation breakdown of ViT (Dosovitskiy et al., 2020) and DeiT (Touvron et al., 2021). $L$ is the number of transformer blocks, $N$ is the number of input tokens (patches), and $M$ is the feature dimensionality. All models take input images of size $224 \times 224$ except ViT-B/384, which uses $384 \times 384$. The softmax-attention layers constitute a fraction (15% or less) of the total compute, whereas fully-connected layers (MLP and projections) spend over 80%.

(a) Score-based downsampling

(b) Proposed downsampling

Figure 2: Score-based downsampling methods (Goyal et al., 2020; Rao et al., 2021) *vs.* our method. In the figure, the x-axis represents the token values (in one dimension), and the y-axis represents their scores. Suppose four tokens are to be selected. (a) Score-based methods select tokens with higher scores. Since the scoring function is continuous, all tokens in the left lobe will be selected, resulting in redundancy and information loss in the right lobe. (b) The proposed method first forms four clusters to approximate the set of tokens, then selects the cluster centers. Thus, the output tokens are a more accurate representation of the original token set than the score-based methods.

softmax-attention (3), QKV projections (4), O projection (2) and MLP (1). As shown in Table 1, in all these vision transformers, the main computational bottleneck is the fully-connected layers that spend over 80% of the total computation. In comparison, softmax-attention only takes less than 15%. Note that we explicitly break down the multi-head attention into the softmax-attention, QKV and O projections, as they have different time complexity, see Table 1. This decomposition reveals that the QKV and O projections spend most of the computations of the multi-head self-attention.

## 3.2 LIMITATIONS OF SCORE-BASED TOKEN DOWNSAMPLING

Existing score-based token downsampling methods like PoWER-BERT and Dynamic-ViT utilize scoring functions to determine the tokens to keep (or prune). They keep tokens with the top-K scores and discard the rest. Since these scoring functions are continuous with limited Lipschitz constants, tokens that are close in the feature space will be assigned similar scores. Therefore, these similar tokens will likely either be all kept or discarded, as illustrated in Figure 2a. As our experiments show, this redundancy (in the kept tokens) and severe information loss (in the pruned tokens) deteriorate the computation-accuracy trade-off of the score-based downsampling methods.

### 3.3 ATTENTION AS A LOW-PASS FILTER

Given a query vector $\boldsymbol{q}$, a set of key vectors $\mathcal{K} = \{\boldsymbol{k}_1, \ldots, \boldsymbol{k}_N\}$, the corresponding value vectors $\mathcal{V} = \{\boldsymbol{v}_1, \ldots, \boldsymbol{v}_N\}$ and a scalar $\alpha > 0$, softmax-attention computes the output via

$$\boldsymbol{o}(\boldsymbol{q}) = \frac{1}{z(\boldsymbol{q})} \sum_{i=1}^{N} \exp(\alpha\, \boldsymbol{q} \cdot \boldsymbol{k}_i)\, \boldsymbol{v}_i \qquad \text{where} \qquad z(\boldsymbol{q}) = \sum_{i=1}^{N} \exp(\alpha\, \boldsymbol{q} \cdot \boldsymbol{k}_i). \tag{6}$$

Note that we write $\boldsymbol{o}(\boldsymbol{q})$ to indicate that the output vector $\boldsymbol{o}$ is a function of the query $\boldsymbol{q}$. If the query vector and all key vectors are normalized to have a fixed $\ell^2$ norm, we can rewrite (6) as

$$\boldsymbol{o}(\boldsymbol{q}) = \frac{1}{z'(\boldsymbol{q})} \sum_{i=1}^{N} \exp\left(-\frac{\alpha}{2}\|\boldsymbol{q} - \boldsymbol{k}_i\|^2\right) \boldsymbol{v}_i = \frac{1}{z'(\boldsymbol{q})} \int \exp\left(-\frac{\alpha}{2}\|\boldsymbol{q} - \boldsymbol{k}\|^2\right) \left(\sum_{i=1}^{N} \delta(\boldsymbol{k} - \boldsymbol{k}_i)\boldsymbol{v}_i\right) \mathrm{d}\boldsymbol{k}$$

$$= \frac{1}{z'(\boldsymbol{q})}\, G\left(\boldsymbol{q}; \frac{1}{\alpha}\right) * S(\boldsymbol{q}; \mathcal{K}, \mathcal{V}) \tag{7}$$

where $*$ represents high-dimensional convolution, $z'(\boldsymbol{q}) = \sum_i \exp\left(-\frac{\alpha}{2}\|\boldsymbol{q} - \boldsymbol{k}_i\|^2\right) = G\left(\boldsymbol{q}; \frac{1}{\alpha}\right) * S(\boldsymbol{q}; \mathcal{K}, 1)$ is the normalization scalar function, $G\left(\boldsymbol{q}; \sigma^2\right) = \exp\left(-\|\boldsymbol{q}\|^2 / 2\sigma^2\right)$ is an isometric Gaussian kernel, and $S(\boldsymbol{q}; \mathcal{K}, \mathcal{V}) = \sum_{i=1}^{N} \delta(\boldsymbol{q} - \boldsymbol{k}_i)\, \boldsymbol{v}_i$ is a high-dimensional sparse signal, which is composed of $N$ delta functions located at $\boldsymbol{k}_i$ with value $\boldsymbol{v}_i$. According to (7), given query vectors $\boldsymbol{q}_1, \ldots, \boldsymbol{q}_N$, there are two conceptual steps to compute softmax-attention:

1. filter $S(\boldsymbol{q}; \mathcal{K}, \mathcal{V})$ with a Gaussian kernel to get $\boldsymbol{o}(\boldsymbol{q})$, and

2. sample $\boldsymbol{o}(\boldsymbol{q})$ at coordinates $\boldsymbol{q}_1, \ldots, \boldsymbol{q}_N$ to get the output vectors $\boldsymbol{o}_1, \ldots, \boldsymbol{o}_N$.

Since Gaussian filtering is low-pass, $\boldsymbol{o}(\boldsymbol{q})$ is a smooth signal. Therefore, the output tokens of the attention layer, $i.e.$, discrete samples of $\boldsymbol{o}(\boldsymbol{q})$, contain similar feature values. So, based on Nyquist-Shannon sampling theorem (Shannon, 1949), there exists redundant information in the output, which is used by our methods to reduce computation without losing much of the important information.

Note that our analysis is based on the normalized query and key vectors, which can be achieved by inserting a normalizing layer before the softmax-attention layer without significantly affecting the performance of a transformer, as shown by Kitaev et al. (2020). It has also been empirically observed by Goyal et al. (2020) and Rao et al. (2021) that even without the normalization, transformers produce tokens with similar values. To demonstrate this, in all our experiments, we use standard multi-head attention *without* normalizing keys and queries. We conduct an ablation study with normalized keys and queries in Appendix F.

## 4 TOKEN POOLING

Pruning tokens inevitably loses information. In this section, we formulate a new token downsampling principle enabling strategical tokens selection that preserves the most information. Based on this principle, we formulate and discuss several Token Pooling algorithms.

Given a set of output tokens $\mathcal{F} = \{\boldsymbol{f}_1, \ldots, \boldsymbol{f}_N\}$ of a transformer block, our goal is to find a smaller set of tokens $\hat{\mathcal{F}} = \{\hat{\boldsymbol{f}}_1, \ldots, \hat{\boldsymbol{f}}_K\}$ that after upsampling minimizes the reconstruction error of $\mathcal{F}$. Specifically, the reconstruction of $u(\boldsymbol{f}_i; \hat{\mathcal{F}})$ of token $\boldsymbol{f}_i$ is computed via interpolating the tokens in $\hat{\mathcal{F}}$. The reconstruction error is then defined as

$$\ell(\mathcal{F}, \hat{\mathcal{F}}) = \sum_{\boldsymbol{f}_i \in \mathcal{F}} \|\boldsymbol{f}_i - u(\boldsymbol{f}_i; \hat{\mathcal{F}})\|^2. \tag{8}$$

To simplify our formulation and reduce computation, we use *nearest-neighbor* interpolation as $u$. As a consequence, the reconstruction error (8) becomes

$$\ell(\mathcal{F}, \hat{\mathcal{F}}) = \sum_{\boldsymbol{f}_i \in F} \min_{\hat{\boldsymbol{f}}_j \in \hat{F}} \|\boldsymbol{f}_i - \hat{\boldsymbol{f}}_j\|^2, \tag{9}$$

which is the asymmetric Chamfer divergence between $\mathcal{F}$ and $\hat{\mathcal{F}}$ (Barrow et al., 1977; Mechrez et al., 2018). The loss (9) can be minimized by the K-Means algorithm, $i.e.$, clustering the tokens in $\mathcal{F}$ into $K$ clusters.

---

**Algorithm 1:** Token Pooling

---

**input** : tokens $\mathcal{F} = \{\boldsymbol{f}_1, \ldots, \boldsymbol{f}_N\}$; target set size $K$; (optional) weights $\mathcal{W} = \{w_1, \ldots, w_N\}$
**output:** downsampled set $\hat{\mathcal{F}} = \{\hat{\boldsymbol{f}}_1, \ldots, \hat{\boldsymbol{f}}_K\}$

1 **if** $k \geq N$ **then return** $F$;

2 initialize cluster centers $\hat{\mathcal{F}}$ to be the $K$ tokens from $\mathcal{F}$ with the highest weights ;

3 **while** *not converged* and *max number of iterations is not reached* **do**

4     **for** $i \in \{1, ..., N\}$ **do** update cluster assignment $z(i) \leftarrow \arg\min_{j=1}^{K} \|\boldsymbol{f}_i - \hat{\boldsymbol{f}}_j\|$ ;

5     **for** $j \in \{1, ..., K\}$ **do** update cluster center $\hat{\boldsymbol{f}}_j$ according to the chosen clustering algorithm, that is either (11), (12), (15) or (18) ;

6 **return** *weighted means of tokens in each cluster*

---

The proposed Token Pooling layer is defined in Algorithm 1. It downsamples input tokens via clustering the tokens and returns the cluster centers (the average of the tokens in a cluster). As we have shown above, this operation directly minimizes the reconstruction error (9) caused by the downsampling. Intuitively, clustering the tokens provides a more accurate and diverse representation of the original set of tokens, compared to the top-K selection used by score-based downsampling methods, as shown in Figure 2b. Note that Token Pooling is robust to the initialization of cluster centers, as shown in Appendix B. Below, we provide details of the clustering algorithms.

**K-Means.** We use the K-Means algorithm to minimize (9) via the following iterations:

$$a(i) \quad \leftarrow \quad \arg\min_{j} \|\boldsymbol{f}_i - \hat{\boldsymbol{f}}_j\| \qquad\qquad \forall i \in \{1, 2, \ldots, N\} \qquad (10)$$

$$\hat{\boldsymbol{f}}_j \quad \leftarrow \quad \sum_{i=1}^{N}[a(i) = j]\boldsymbol{f}_i \Big/ \sum_{i=1}^{N}[a(i) = j] \qquad \forall j \in \{1, 2, \ldots, K\} \qquad (11)$$

where $[\,]$ is the Iverson bracket and $a$ is the cluster assignment function. The overall algorithm complexity is $\mathcal{O}(TKNM)$ where $T$ is the number of iterations. The vast majority of the computation is spent on the repetitive evaluation of the distances between tokens and centroids in step (10).

**K-Medoids.** We can use the more efficient K-Medoids algorithm by replacing step (11) with:

$$n(j) \quad \leftarrow \quad \arg\min_{i:a(i)=j} \sum_{i':a(i')=j} \|\boldsymbol{f}_i - \boldsymbol{f}_{i'}\|^2 \qquad \text{and} \qquad \hat{\boldsymbol{f}}_j \quad \leftarrow \quad \boldsymbol{f}_{n(j)}. \qquad (12)$$

These steps minimize objective (9) under the *medoid constraint*: $\hat{\mathcal{F}} \subseteq \mathcal{F}$.

The advantage of the K-Medoids algorithm is its time complexity $\mathcal{O}(TKN + N^2M)$, which is substantially lower in practice as we only compute the distances between tokens once. In our experience, it requires less than 5 iterations to converge. Apart from the distance matrix computation, the cost of the K-Medoids algorithm is negligible when compared with the cost of the other layers.

**Weighted clustering.** Reconstruction error (9) treats every token equally; however, each token contributes differently to the final output of an attention layer. Thus, we also consider a weighted reconstruction error: $\ell(\mathcal{F}, \hat{\mathcal{F}}; \boldsymbol{w}) = \sum_{\boldsymbol{f}_i \in F} \min_{\hat{\boldsymbol{f}}_j \in \hat{F}} w_i \|\boldsymbol{f}_i - \hat{\boldsymbol{f}}_j\|^2$ where $\boldsymbol{w} = [w_1, \ldots, w_N]$ are the positive weights corresponding to the individual tokens in $\mathcal{F}$. Appendix A details the clustering algorithms for the weighted case. A good choice of the weights is the significance scores (5), *i.e.*, $w_i = s_i$. The significance score identifies the tokens that influence the current transformer block most and thus should be approximated more precisely.

Note, when applying Token Pooling, it is important to keep special tokens like the classification **NEW** token. If spatial locations on an image are need, *e.g.*, in the case of semantic segmentation or pose estimation, techniques like (Wu et al., 2020) can be utilized to project tokens back to the image grid.

## 5 EXPERIMENTS

Our implementation is based on DeiT (Touvron et al., 2021) where we insert downsampling layers after each transformer block. To evaluate the pure effect of downsampling, we keep all meta-parameters of DeiT, including the feature dimensionality, network depth, learning rate schedule, *etc*.

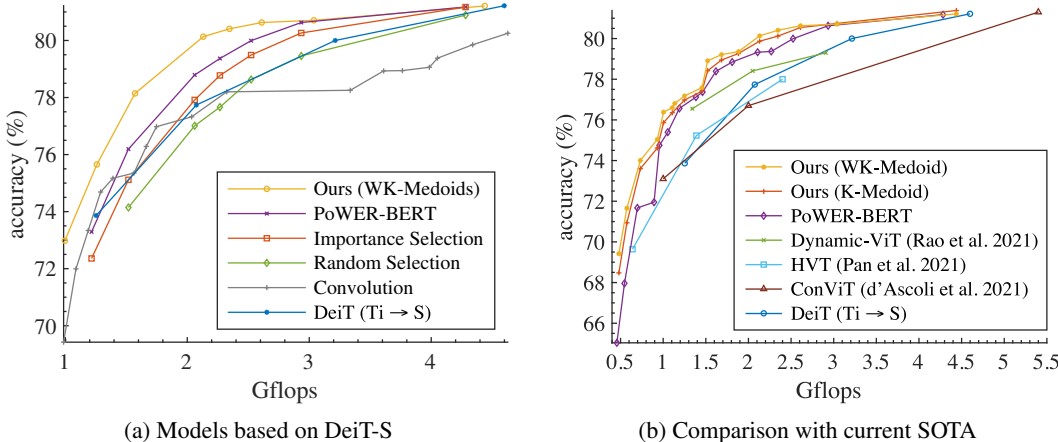

(a) Models based on DeiT-S    (b) Comparison with current SOTA

Figure 3: **Main results.** (a) shows the accuracy when we apply different downsampling methods to DeiT-S. More is in Appendix G. (b) shows a comparison between the proposed method with the state-of-the-art downsampling methods. The results of our method and PoWER-BERT are acquired by varying $K$ and the base architecture among DeiT-Ti, DeiT-e252, DeiT-e318, and DeiT-S.

We do not use knowledge distillation. We use ImageNet1k (Russakovsky et al., 2015). Our cost and performance metrics are flops and top-1 accuracy. Appendix H reports throughput of our models.   **NEW**

**Methods.** We evaluate the following token downsampling methods:

1. *Convolution* downsampling. We implement uniform grid-downsampling via convolution with stride 2, *i.e.*, the tokens corresponding to the adjacent image patches are concatenated and mapped to $\mathbb{R}^M$. This design is used in Liu et al. (2021); Heo et al. (2021). See details in Appendix D.

2. *PoWER-BERT.* We implement PoWER-BERT on DeiT, following Goyal et al. (2020).

3. *Random selection*, a simple baseline randomly selecting $K$ tokens without replacement.

4. *Importance selection* chooses $K$ tokens by drawing from a categorical distribution without replacement with probabilities proportional to the significance score (5) of each token.

5. *Token Pooling* use K-Means or K-Medoids algorithms, or their weighted versions, WK-Means or WK-Medoids, respectively. The weights are the significance scores (5).

**Selection of $K = [K_1, \ldots, K_L]$.** To fairly compare our Token Pooling with PoWER-BERT and other baselines, all methods (except convolution downsampling) use the same number of retained tokens for downsampling layers. Appendix C details the selection of $K$.

**Training protocol.** (1) A base DeiT model (*e.g.*, DeiT-S) is trained using the original training (Touvron et al., 2021). (2) We then finetune the base DeiT model using the second stage of PoWER-BERT's training and acquire $K$. (3) We further finetune the downsampling methods using $K$. (4) We also finetune the base DeiT model. Our protocol ensures a fair comparison such that all of the models are trained with the same number of iterations, the same learning rate schedule, *etc*.

## 5.1 MAIN RESULTS

First, we apply different downsampling methods on DeiT-S. As shown in Figure 3a, random selection achieves a similar trade-off as lowering feature dimensionality $M$. While convolution with stride is better than adjusting $M$ at the low-compute regime, it fails in high-compute regimes. Importance selection improves upon random selection but is still outperformed by PoWER-BERT. Our Token Pooling (with weighted K-Medoids) achieves the best trade-off in all regimes.

Next, we apply Token Pooling to DeiT models with different $M$ (DeiT-Ti, DeiT-e252, DeiT-e318, and DeiT-S). Figure 4 shows trade-off curves for each DeiT model. Token Pooling enables each of the models to achieve a better computation-accuracy trade-off than simply varying the feature dimensionality $M$. For each computational budget, we find the combination of $M$ and $K$ that gives the highest accuracy. The best balance is achieved by applying Token Pooling and selecting the best

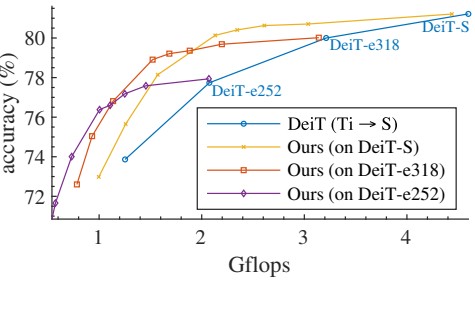

Figure 4: The figure shows the results when we apply Token Pooling to various DeiT architectures. Token Pooling consistently improves the computation-accuracy trade-off for all evaluated architectures. By utilizing both Token Pooling and architecture search, we can further improve the accuracy at a given flops budget. For example, at 1 Gflop, we should use Token Pooling on DeiT-e252 instead of DeiT-S.

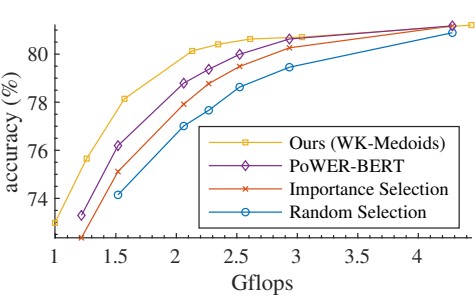

(a) Ours *vs.* methods using significance score

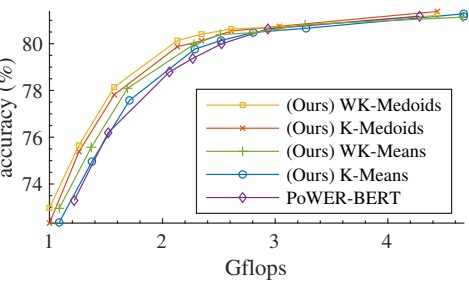

(b) Variants of Token Pooling

Figure 5: Ablation studies of (a) downsampling methods using significance score, and (b) proposed Token Pooling using different clustering algorithms. The base model is DeiT-S for all methods.

$M$. We do the same for PoWER-BERT. Figure 3b shows the results of the proposed Token Pooling and state-of-the-art methods. We cite the results of Rao et al. (2021), d'Ascoli et al. (2021), and Pan et al. (2021). Our Token Pooling achieves the best accuracy across all evaluated compute regimes.

Finally, we compare the best computation-accuracy trade-off achieved by Token Pooling (with WK-Medoids and varying $M$) with standard DeiT models in Figure 1b. As can be seen, utilizing Token Pooling, we significantly improve the computation costs of DeiT-Ti by 42% and improve the top-1 accuracy by 3.3 points at the same flops. Similar benefits can be seen on DeiT-e252 and DeiT-e318.

## 5.2 ABLATION STUDIES

Figure 5a compares methods utilizing significance scores. As can be seen, using importance selection improves upon the simple random selection. By minimizing the reconstruction error (9), our method achieves better cost-accuracy trade-off. Figure 5b evaluates Token Pooling with different clustering algorithms. Weighted versions outperform regular versions of K-Means and K-Medoids. Due to the higher time complexity, K-Means is outperformed by K-Medoids (the curves are shifted toward the right). See the metrics and flops used by clustering in table format in Appendix G. All Token Pooling variants outperform the baseline, demonstrating the effectiveness of our method.

More ablation studies are in the appendices: clustering initialization (Appendix B), convolution downsampling (D), Token Pooling without finetuning (E), results with normalized keys and queries (F), and detailed information ($K$, flops, accuracy, clustering cost) of our models (G).

## 6 CONCLUSIONS

This paper provides two insights of vision transformers: first, their computational bottleneck is the fully-connected layers, and second, attention layers generate redundant representations due to the connection to Gaussian filtering. Token Pooling, our novel nonuniform data-aware downsampling operator, utilizes these insights and significantly improves the computation-accuracy balance of DeiT, compared to existing downsampling techniques. We believe that Token Pooling is a timely development and can be used with other techniques (*e.g.*, architecture search, quantization) to spur the design of efficient vision transformers.

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

# A    WEIGHTED CLUSTERING ALGORITHMS

**Weighted K-Means**    minimizes the following objective w.r.t. $\hat{\mathcal{F}} = \{\hat{\boldsymbol{f}}_1, \ldots, \hat{\boldsymbol{f}}_K\} \subset \mathbb{R}^M$:

$$\ell(\mathcal{F}, \hat{\mathcal{F}}) = \sum_{\boldsymbol{f}_i \in F} \min_{\hat{\boldsymbol{f}}_j \in \hat{F}} w_i \|\boldsymbol{f}_i - \hat{\boldsymbol{f}}_j\|^2 \tag{13}$$

The extension of the K-Means algorithm to the weighted case iterates the following steps:

$$a(i) \quad \leftarrow \quad \arg\min_j \|\boldsymbol{f}_i - \hat{\boldsymbol{f}}_j\| \qquad\qquad \forall i \in \{1, 2, \ldots, N\}, \tag{14}$$

$$\hat{\boldsymbol{f}}_j \quad \leftarrow \quad \frac{\sum_{i=1}^{N}[a(i) = j] w_i \boldsymbol{f}_i}{\sum_{i=1}^{N}[a(i) = j] w_i} \qquad\qquad \forall j \in \{1, 2, \ldots, K\}., \tag{15}$$

**Weighted K-Medoids**    optimizes objective (13) under the medoid constraint $\hat{\mathcal{F}} \subset \mathcal{F}$:

$$a(i) \quad \leftarrow \quad \arg\min_j \|\boldsymbol{f}_i - \hat{\boldsymbol{f}}_j\| \qquad\qquad \forall i \in \{1, 2, \ldots, N\}, \tag{16}$$

$$n(j) \quad \leftarrow \quad \arg\min_{i: z(i)=j} \sum_{i': a(i')=j} \|\boldsymbol{f}_i - \boldsymbol{f}_{i'}\|^2 \qquad\qquad \forall j \in \{1, 2, \ldots, K\}, \tag{17}$$

$$\hat{\boldsymbol{f}}_j \quad \leftarrow \quad \boldsymbol{f}_{n(j)}. \tag{18}$$

# B    CLUSTERING INITIALIZATION ABLATIONS

We examine the effect of the cluster center initialization. We compare our default initialization, which uses the tokens with top-K significance scores as initial cluster centers, with random initialization, which randomly selects tokens as initial cluster centers. As shown in Figure 6, Token Pooling is robust to the initialization methods.

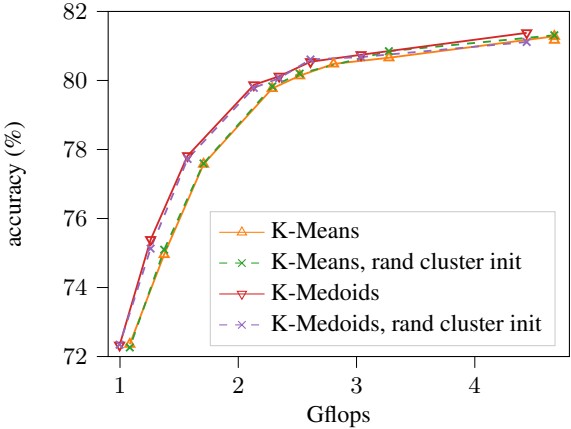

Figure 6: Default initialization *vs.* random initialization. Our Token Pooling is robust to initialization of clustering algorithms. The default initialization is top-K w.r.t. significance score, see Algorithm 1.

# C    TRAINING DETAILS & HYPERPARAMETERS

A Token Pooling layer has one parameter—the number of the output tokens $K$ (or equivalently the downsampling ratio)—which is the same as downsampling layers like max or average pooling. For max or average pooling layers, the sizes of the kernels and strides play the same role as $K$. We use the method proposed by Goyal et al. (2020) to automatically select $K$s of all Token Pooling layers (see below for details). All other hyperparameters are inherited from DeiT (Touvron et al., 2021). **NEW**

To fairly compare PoWER-BERT and the baseline methods with the proposed Token Pooling, all methods (except convolution downsampling) use the same target number of tokens for downsampling layers after each transformer block. Specifically, we run the second stage of the PoWER-BERT training (Goyal et al., 2020) for 30 epochs with various values of the token-selection weight parameter $\lambda$ producing a family of models. The single parameter $\lambda$ controls the overall efficiency of a model, and the numbers of retained tokens of all Token Pooling layers are selected automatically based on the choice of $\lambda$. Each of the resulted models has a different number of retained tokens at each of its $L$ transformer blocks: $\boldsymbol{K} = (K_1, \ldots, K_L)$. Appendix G lists all combinations of automatically determined $\boldsymbol{K}$. We then finetune these models using the third (last) stage of PoWER-BERT. Note that we apply the same process for PoWER-BERT, random selection, importance selection, and our Token Pooling, and we use the same $\boldsymbol{K}$ when comparing these methods.

We find that the DeiT models provided by Touvron et al. (2021) are under-fit, and their accuracy improves with additional training, see Figure 7. After the standard DeiT training, we restart the training. This ensures that downsampling models and DeiT with a similar number of training steps.

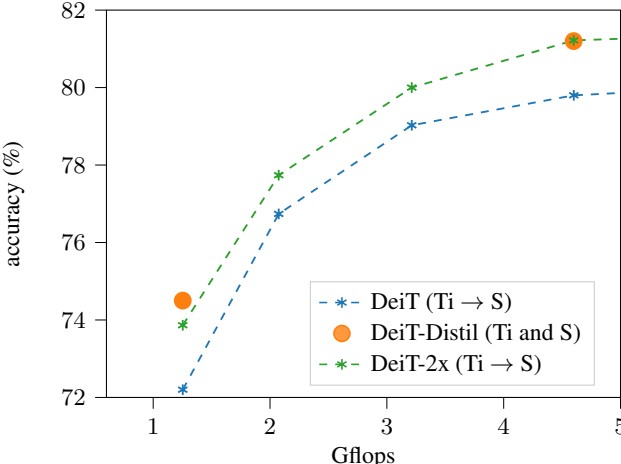

Figure 7: This figure shows the results of the pretrained DeiT models provided by Touvron et al. (2021) (*DeiT*) and the DeiT models trained with our protocol (*DeiT-2x*). Our training protocal uses the same hyper-parameters provided by Touvron et al. (2021), but after the model is trained, we finetune the model using the same hyper-parameters (*i.e.*, restart the learning rate schedule). We also show *DeiT-Distil* results (cited from (Touvron et al., 2021)), which use knowledge distillation.

## D  CONVOLUTION DOWNSAMPLING

As mentioned in the main paper, we enumerate combinations of layers to insert the convolution downsampling layer. We use 2x2 convolution with stride 2 like in Liu et al. (2021). To keep the feature dimensionality of DeiT the same (and evaluate the pure effect of the downsampling layers), the output feature dimensionality is the same as the input. With 196 tokens in DeiT-S model, we can include no more than 3 convolution downsampling layers as each layer reduces the number of token by a factor of 4. When using 3 layers at depths $l_1 < l_2 < l_3$, we restrict $l_2 - l_1 = l_3 - l_2$. With these constraints, we enumerate all possible downsampling configurations. Each of the combinations produces a model with a different computation-accuracy trade-off, and we report the Pareto front, *i.e.*, the best accuracy these models achieve at a given flop. See Figure 8.

Note that the convolution downsampling layer is equivalent to the patch-merging layer used by Liu   **NEW**
et al. (2021), Heo et al. (2021), and Wang et al. (2021) (except that we keep the output feature dimension the same to compare fairly with DeiT). It is also a generalized version of the commonly used average pooling layer. Average pooling can be implemented by convolving the feature map with a constant kernel of size $n \times n$ containing $\frac{1}{n^2}$ and subsample (stride) every $n$ pixels. By learning its kernel, the convolution downsampling layer is able to find a kernel suitable for the downsampling and the current task than average pooling.

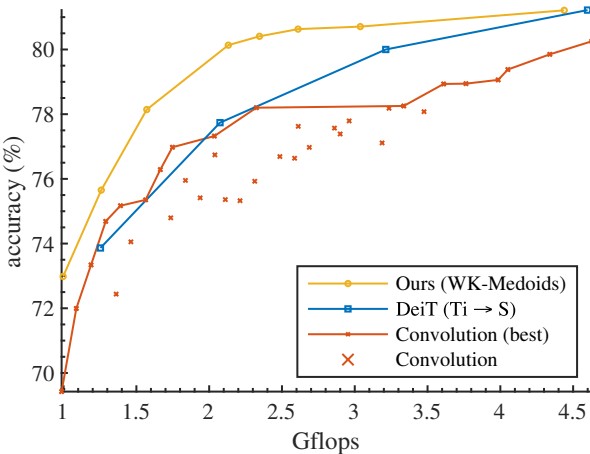

Figure 8: Results of convolution downsampling

## E  RESULTS WITHOUT FINETUNING

Since Token Pooling minimizes the reconstruction error due to token downsampling, in this section, we evaluate the performance when we insert Token Pooling layers into a pre-trained model *without* finetuning. Figure 9 shows the results when we directly insert Token Pooling layers (using the same downsampling ratios in Figure 5b) into a pretrained DeiT-S. As can be seen, minimizing the reconstruction error, Token Pooling preserves information that enables the model to retain accuracy during token downsampling.

In Figure 9, we also show the results when we replace tokens to their cluster centers (WK-Means, carry, no finetuning and WK-Medoids, carry, no finetuning). Specifically, in addition to outputting $K$ cluster centers, we count the number of tokens assigned to a cluster center and *carry* the count when we compute softmax-attention in the next transformer blocks. This operation preserves the attention weights, and since the models are not trained after inserting Token Pooling layers, it preserves the most accuracy. In practice, when the models are trained with Token Pooling layers, the carry operation is not important.

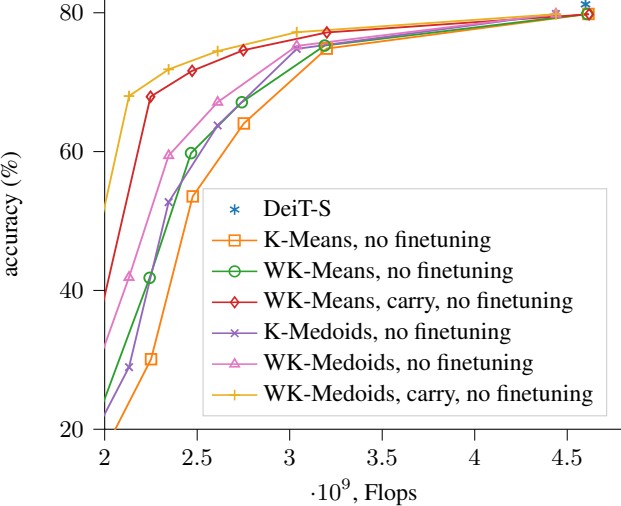

Figure 9: The figure shows the performance results when we insert Token Pooling layers into a pretrained DeiT-S *without* finetuning the model (*i.e.*, skip step 3 of the training protocol in Section 5).

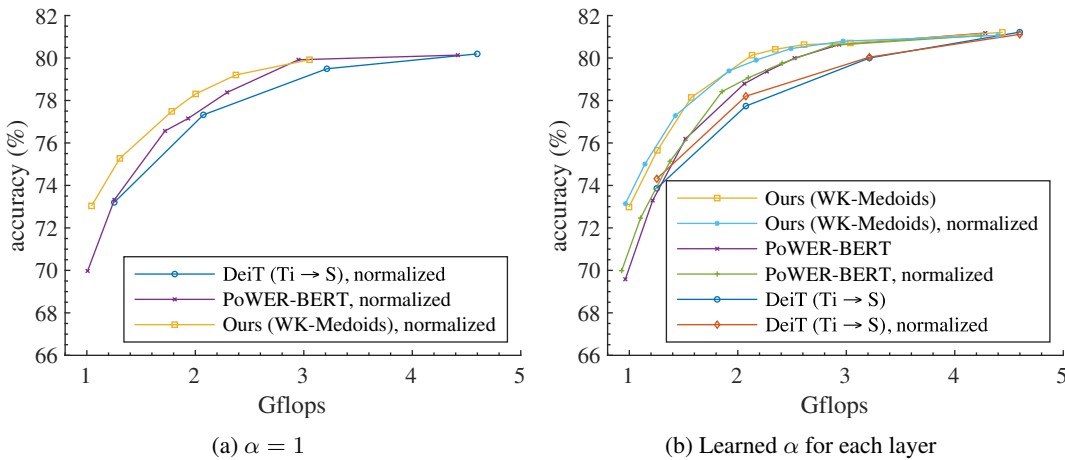

Figure 10: Results of models using normalized key and query vectors with (a) $\alpha = 1$ and (b) learned $\alpha$ in (6). The base model architecture is DeiT-S.

## F   NORMALIZED KEY AND QUERY VECTORS

Our analysis of softmax-attention in §3.3 assumes the query and the key vectors are normalized to have a constant $\ell^2$ norm. It has been observed by Kitaev et al. (2020) that normalizing key and query vectors does not change the performance of a transformer. Thus, for all experiments, we train models *without* the norm normalization. In this section, we verify this observation by training various DeiT, PoWER-BERT, and Token Pooling *with* normalized keys and queries. We found that the scalar $\alpha$ in the softmax attention layer (6) can affect the performance of a transformer with normalized keys and queries. As can be seen in Figure 10a, setting the scalar $\alpha = 1$ slightly deteriorates the performance of the model. Instead of using a fixed $\alpha$, we let the model learn the $\alpha$ for each layer. As can be seen in Figure 10b, learning $\alpha$ enables the resulting models to achieve similar cost-accuracy trade-off as the standard (unnormalized) models. With or without the normalization and the learnable $\alpha$, the proposed Token Pooling significantly improves the cost-accuracy trade-off of DeiT and outperforms PoWER-BERT.

## G   CLUSTERING ALGORITHM ABLATIONS

Tables 2–5 detail the results of PoWER-BERT and Token Pooling on the DeiT architectures that we tested (DeiT-S, DeiT-e318, and DeiT-e252). Figure 11 shows the ablation results with different backbone models. Table 2 details the results of the best cost-accuracy trade-off achieved by the proposed Token Pooling (using K-Medoids and WK-Medoids) and PoWER-BERT via varying token sparsity and feature dimensionality of DeiT.

Apart from the standard K-Means and K-Medoids, other clustering approaches could be used. Many methods are not suitable due to efficiency constraints. For example, normalized cut (Shi & Malik, 2000) uses expensive spectral methods. One prerequisite of using K-Means and K-Medoids is the number of clusters, $K$. While selecting $K$ for each of the layers may be tedious and difficult, one can choose $K$ via computational budgets and heuristics. In this work, we use the automatic search procedure proposed by Goyal et al. (2020) to determine $K$, see Appendix C.

Since both K-Means and K-Medoids require specifying the number of clusters $K$ in advance, one may consider using methods automatically determining $K$. Nevertheless, such methods typically have other parameters, which are less interpretable than $K$. For example, mean-shift (Cheng, 1995) or quick-shift (Vedaldi & Soatto, 2008) require specifying the kernel size. From our experience, determining these parameters is challenging. Also, since the number of clusters (and hence the number of output tokens) is determined on the fly during inference, the computational requirement can fluctuate, making deployment of these models difficult.

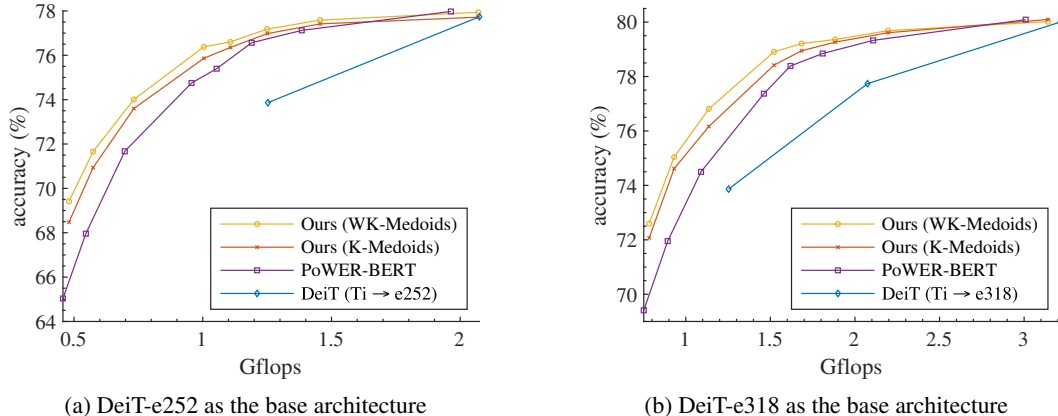

(a) DeiT-e252 as the base architecture      (b) DeiT-e318 as the base architecture

Figure 11: This figure compares the cost-accuracy curves of Token Pooling with PoWER-BERT using (a) DeiT-e252 and (b) DeiT-e318 as the base architectures.

| DeiT (Ti → S) | | PoWER-BERT | | Token Pooling (K-Medoids) | | Token Pooling (WK-Medoids) | |
|---|---|---|---|---|---|---|---|
| Gflops | Accuracy (%) | Gflops | Accuracy (%) | Gflops | Accuracy (%) | Gflops | Accuracy (%) |
| - | - | 0.46 | 65.0 | 0.48 | 68.5 | 0.48 | **69.4** |
| - | - | 0.55 | 68.0 | 0.57 | 70.9 | 0.57 | **71.7** |
| - | - | 0.70 | 71.7 | 0.73 | 73.6 | 0.73 | **74.0** |
| - | - | 0.89 | 72.0 | 0.93 | 74.6 | 0.93 | **75.0** |
| - | - | 0.96 | 74.8 | 1.00 | 75.9 | 1.00 | **76.4** |
| - | - | 1.05 | 75.4 | 1.11 | 76.4 | 1.11 | **76.6** |
| 1.25 | 73.9 | 1.19 | 76.6 | 1.25 | 77.0 | 1.25 | **77.2** |
| - | - | 1.38 | 77.1 | 1.46 | 77.4 | 1.46 | **77.6** |
| - | - | 1.46 | 77.4 | 1.52 | 78.4 | 1.52 | **78.9** |
| - | - | 1.62 | 78.4 | 1.68 | 78.9 | 1.68 | **79.2** |
| 2.07 | 77.7 | 1.81 | 78.8 | 1.88 | 79.3 | 1.88 | **79.4** |
| - | - | 2.11 | 79.3 | 2.13 | 79.9 | 2.13 | **80.1** |
| - | - | 2.27 | 79.4 | 2.35 | 80.1 | 2.35 | **80.4** |
| - | - | 2.52 | 80.0 | 2.61 | 80.5 | 2.61 | **80.6** |
| 3.21 | 80.0 | 2.93 | 80.6 | 3.04 | **80.7** | 3.04 | **80.7** |
| 4.60 | 81.2 | 4.28 | 81.2 | 4.44 | **81.4** | 4.44 | 81.2 |

Table 2: Best cost-accuracy trade-off achieved by PoWER-BERT and the proposed Token Pooling via varying sparsity level and feature dimensionality.

| clustering method | Weighting | ImageNet Accuracy | GFlops |
|---|---|---|---|
| Sparsity level 0: $\boldsymbol{K} = [196, 196, 195, 194, 189, 180, 173, 173, 173, 173, 173, 173]$ | | | |
| PoWER-BERT | N/A | 81.2 | 4.3 |
| K-Means | | 81.3 | 4.7 (+0.4) |
| | ✓ | 81.1 | 4.7 (+0.4) |
| K-Medoids | | **81.4** | 4.4 (+0.1) |
| | ✓ | 81.2 | 4.4 (+0.1) |
| Sparsity level 1: $\boldsymbol{K} = [196, 195, 193, 188, 169, 140, 121, 110, 73, 38, 7, 0]$ | | | |
| PoWER-BERT | N/A | 80.6 | 2.9 |
| K-Means | | 80.7 | 3.3 (+0.4) |
| | ✓ | **80.8** | 3.3 (+0.4) |
| K-Medoids | | 80.7 | 3.0 (+0.1) |
| | ✓ | 80.7 | 3.0 (+0.1) |
| Sparsity level 2: $\boldsymbol{K} = [196, 195, 190, 177, 141, 108, 84, 69, 35, 18, 3, 0]$ | | | |
| PoWER-BERT | N/A | 80.0 | 2.5 |
| K-Means | | 80.5 | 2.8 (+0.3) |
| | ✓ | 80.5 | 2.8 (+0.3) |
| K-Medoids | | 80.5 | 2.6 (+0.1) |
| | ✓ | **80.6** | 2.6 (+0.1) |
| Sparsity level 3: $\boldsymbol{K} = [196, 194, 187, 163, 118, 85, 58, 47, 20, 12, 2, 0]$ | | | |
| PoWER-BERT | N/A | 79.4 | 2.3 |
| K-Means | | 80.1 | 2.5 (+0.2) |
| | ✓ | **80.4** | 2.5 (+0.2) |
| K-Medoids | | 80.1 | 2.3 (+0.08) |
| | ✓ | **80.4** | 2.3 (+0.08) |
| Sparsity level 4: $\boldsymbol{K} = [196, 193, 179, 142, 97, 64, 46, 34, 13, 9, 1, 0]$ | | | |
| PoWER-BERT | N/A | 78.8 | 2.1 |
| K-Means | | 79.8 | 2.3 (+0.2) |
| | ✓ | 80.0 | 2.3 (+0.2) |
| K-Medoids | | 79.9 | 2.1 (+0.07) |
| | ✓ | **80.1** | 2.1 (+0.07) |
| Sparsity level 5: $\boldsymbol{K} = [194, 183, 142, 89, 41, 20, 10, 7, 0, 0, 0, 0]$ | | | |
| PoWER-BERT | N/A | 76.2 | 1.5 |
| K-Means | | 77.6 | 1.7 (+0.2) |
| | ✓ | **78.1** | 1.7 (+0.2) |
| K-Medoids | | 77.8 | 1.6 (+0.05) |
| | ✓ | **78.1** | 1.6 (+0.05) |
| Sparsity level 6: $\boldsymbol{K} = [186, 162, 102, 56, 13, 4, 2, 2, 0, 0, 0, 0]$ | | | |
| PoWER-BERT | N/A | 73.3 | 1.2 |
| K-Means | | 75.0 | 1.4 (+0.2) |
| | ✓ | 75.6 | 1.4 (+0.2) |
| K-Medoids | | 75.4 | 1.3 (+0.04) |
| | ✓ | **75.7** | 1.3 (+0.04) |
| Sparsity level 7: $\boldsymbol{K} = [162, 129, 66, 33, 4, 1, 1, 0, 0, 0, 0, 0]$ | | | |
| PoWER-BERT | N/A | 69.6 | 1.0 |
| K-Means | | 72.4 | 1.1 (+0.1) |
| | ✓ | **73.0** | 1.1 (+0.1) |
| K-Medoids | | 72.3 | 1.0 (+0.03) |
| | ✓ | **73.0** | 1.0 (+0.03) |

Table 3: Results of applying Token Pooling and PoWER-BERT on **DeiT-S** model. The models are grouped by $\boldsymbol{K}$ described in Appendix C. The integer list denotes the maximal number of tokens retained after each transformer block. These numbers do not take into account the classification token, which is always retained. Thus, "0" means that only the classification token remains. Additional flops (denoted by the parentheses) are due to clustering.

| clustering method | Weighting | ImageNet Accuracy | GFlops |
|---|---|---|---|
| Sparsity level 0: $K = [196, 196, 196, 194, 192, 184, 181, 181, 170, 170, 170, 170]$ | | | |
| PoWER-BERT | N/A | 80.1 | 3.0 |
| K-Medoids | | **80.1** | 3.1 (+0.1) |
| | ✓ | 80.0 | 3.1 (+0.1) |
| Sparsity level 1: $K = [196, 195, 195, 190, 171, 147, 132, 122, 66, 43, 15, 0]$ | | | |
| PoWER-BERT | N/A | 79.3 | 2.1 |
| K-Medoids | | 79.6 | 2.2 (+0.1) |
| | ✓ | **79.7** | 2.2 (+0.1) |
| Sparsity level 2: $K = [196, 195, 193, 180, 143, 109, 92, 77, 40, 21, 4, 0]$ | | | |
| PoWER-BERT | N/A | 78.8 | 1.8 |
| K-Medoids | | 79.3 | 1.9 (+0.09) |
| | ✓ | **79.4** | 1.9 (+0.09) |
| Sparsity level 3: $K = [196, 195, 190, 165, 119, 84, 65, 50, 26, 14, 3, 0]$ | | | |
| PoWER-BERT | N/A | 78.3 | 1.6 |
| K-Medoids | | 78.9 | 1.7 (+0.07) |
| | ✓ | **79.2** | 1.7 (+0.07) |
| Sparsity level 4: $K = [196, 194, 184, 149, 97, 65, 47, 33, 12, 9, 2, 0]$ | | | |
| PoWER-BERT | N/A | 77.4 | 1.5 |
| K-Medoids | | 78.4 | 1.5 (+0.06) |
| | ✓ | **78.9** | 1.5 (+0.06) |
| Sparsity level 5: $K = [196, 186, 153, 93, 40, 15, 12, 9, 0, 0, 0, 0]$ | | | |
| PoWER-BERT | N/A | 74.5 | 1.1 |
| K-Medoids | | 76.2 | 1.1 (+0.05) |
| | ✓ | **76.8** | 1.1 (+0.05) |
| Sparsity level 6: $K = [193, 173, 109, 52, 16, 4, 4, 4, 0, 0, 0, 0]$ | | | |
| PoWER-BERT | N/A | 72.0 | 0.89 |
| K-Medoids | | 74.6 | 0.93 (+0.04) |
| | ✓ | **75.0** | 0.93 (+0.04) |
| Sparsity level 7: $K = [183, 145, 80, 33, 5, 1, 1, 2, 0, 0, 0, 0]$ | | | |
| PoWER-BERT | N/A | 69.4 | 0.75 |
| K-Medoids | | 72.1 | 0.78 (+0.03) |
| | ✓ | **72.6** | 0.78 (+0.03) |

Table 4: Results of applying Token Pooling and PoWER-BERT on **DeiT-e318** model. The models are grouped by $K$ described in Appendix C. The integer list denotes the maximal number of tokens retained after each transformer block. These numbers do not take into account the classification token, which is always retained. Thus, "0" means that only the classification token remains. Additional flops (denoted by the parentheses) are due to clustering.

| clustering method | Weighting | ImageNet Accuracy | GFlops |
|---|---|---|---|
| Sparsity level 0: $K = [196, 196, 196, 195, 193, 189, 177, 177, 177, 177, 177, 177]$ | | | |
| PoWER-BERT | N/A | **78.0** | 2.0 |
| K-Medoids | | 77.7 | 2.1 (+0.1) |
| | ✓ | 77.9 | 2.1 (+0.1) |
| Sparsity level 1: $K = [196, 196, 195, 189, 176, 161, 123, 122, 76, 48, 17, 0]$ | | | |
| PoWER-BERT | N/A | 77.1 | 1.4 |
| K-Medoids | | 77.4 | 1.5 (+0.07) |
| | ✓ | **77.6** | 1.5 (+0.07) |
| Sparsity level 2: $K = [196, 196, 193, 179, 154, 123, 88, 79, 39, 22, 5, 0]$, | | | |
| PoWER-BERT | N/A | 76.6 | 1.2 |
| K-Medoids | | 77.0 | 1.3 (+0.06) |
| | ✓ | **77.2** | 1.3 (+0.06) |
| Sparsity level 3: $K = [196, 195, 188, 167, 131, 96, 63, 52, 13, 11, 2, 0]$, | | | |
| PoWER-BERT | N/A | 75.4 | 1.1 |
| K-Medoids | | 76.4 | 1.1 (+0.05) |
| | ✓ | **76.6** | 1.1 (+0.05) |
| Sparsity level 4: $K = [196, 194, 181, 147, 111, 75, 48, 36, 5, 7, 2, 0]$ | | | |
| PoWER-BERT | N/A | 74.8 | 0.96 |
| K-Medoids | | 75.9 | 1.0 (+0.04) |
| | ✓ | **76.4** | 1.0 (+0.04) |
| Sparsity level 5: $K = [196, 187, 129, 88, 54, 20, 10, 10, 0, 1, 0, 0]$ | | | |
| PoWER-BERT | N/A | 71.7 | 0.70 |
| K-Medoids | | 73.6 | 0.73 (+0.03) |
| | ✓ | **74.0** | 0.73 (+0.03) |
| Sparsity level 6: $K = [194, 163, 83, 44, 22, 6, 1, 3, 0, 0, 0, 0]$ | | | |
| PoWER-BERT | N/A | 68.0 | 0.55 |
| K-Medoids | | 70.9 | 0.57 (+0.02) |
| | ✓ | **71.7** | 0.57 (+0.02) |
| Sparsity level 7: $K = [188, 122, 58, 28, 12, 2, 0, 2, 0, 0, 0, 0]$ | | | |
| PoWER-BERT | N/A | 65.0 | 0.46 |
| K-Medoids | | 68.5 | 0.48 (+0.02) |
| | ✓ | **69.4** | 0.48 (+0.02) |

Table 5: Results of applying Token Pooling and PoWER-BERT on **DeiT-e252** model. The models are grouped by $K$ described in Appendix C. The integer list denotes the maximal number of tokens retained after each transformer block. These numbers do not take into account the classification token, which is always retained. Thus, "0" means that only the classification token remains. Additional flops (denoted by the parentheses) are due to clustering.

## H  THROUGHPUT

**NEW**

Throughput reflects the actual speed during inference time, and it is often measured by the number of processed images per second (fps). As a result, the value of throughput highly depends on the specific hardware, implementation quality, and the workload and the state (*e.g.*, temperature) of the machines. Since it is usually difficult to control all these factors to have a fair comparison with other papers, in the main paper, we choose to report the theoretical computational cost (*i.e.*, flops) that is known to be highly correlated with the energy consumption on device.

In Figure 12, we provide the throughput of our models, served as a reference on how our models actually perform given our implementation and the type of GPU used. Note that our PyTorch implementation of the clustering algorithms can be significantly improved, for example, via implementing as a CUDA kernel. To determine the throughput, we run the model inference several times with different batch sizes. We report the average throughput of 30 different runs using the best batch size. As can be seen, with the same accuracy, using Token Pooling increases the throughput of DeiT-e318 by 25% (1640 *vs.* 1310 fps), of DeiT-e252 by 22% (2190 *vs.* 1791 fps), and of DeiT-Ti by 14% (3220 *vs.* 2834 fps).

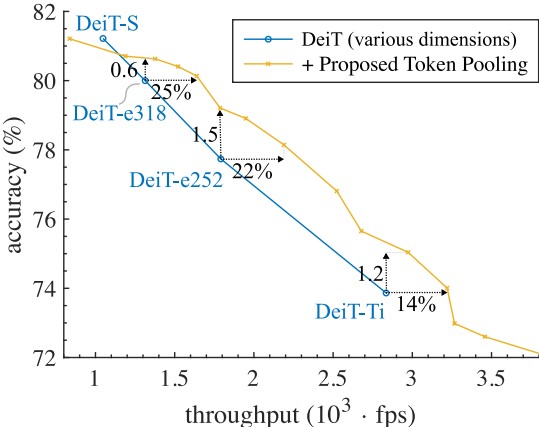

Figure 12: Throughput of our PyTorch implementation. Note that our implementation has not been optimized for the throughput. The throughput is measured in frames (images) per second (fps). These numbers are measured on a single Nvidia V100 GPU. As can be seen, despite our non-optimized code, Token Pooling significantly improves both the flops (see Figure 1b) and throughput of DeiT models.

