# OpenReview forum: "Token Pooling in Vision Transformers"
_ICLR.cc/2022/Conference — ICLR 2022 Submitted_

### Official Review · Reviewer_E53C · 2021-11-01

**Correctness:** 4
**Technical Novelty And Significance:** 3
**Empirical Novelty And Significance:** 3
**Recommendation:** 5
**Confidence:** 4

**Main Review:**

Strength:
+ The submission is technically sound. In Section 3.3, the analysis on attention as a low-pass filter is interesting and makes sense. The K-Means based token down-sampling method is straightforward.
+ The submission is clearly written and well-organized.

Weakness:
- The K-Means clustering method for efficient transformers has been introduced in [1], which narrows the contribution of the paper.
- According to Table 2, 3 and 4, the proposed framework is sensitive to the selection of hyperparameters. It seems that the method is tricky and heavily relies on manual tuning.
- The method is only evaluated on the DeiT. Can it apply to other vision transformers, e.g., PVT and Swin?
- Reducing the number of queries is unfriendly to apply to downstream tasks.
- It would be better to report the actual throughput and compare the proposed method with the previous dynamic vision transformers, such as [2] and [3].

[1] Roy A, Saffar M, Vaswani A, et al. Efficient content-based sparse attention with routing transformers[J]. Transactions of the Association for Computational Linguistics, 2021, 9: 53-68.

[2] Yongming Rao, Wenliang Zhao, Benlin Liu, Jiwen Lu, Jie Zhou, and Cho-Jui Hsieh. DynamicViT: Efﬁcient vision transformers with dynamic token sparsiﬁcation. ArXiv, abs/2106.02034, 2021.

[3] Wang Y, Huang R, Song S, et al. Not All Images are Worth 16x16 Words: Dynamic Vision Transformers with Adaptive Sequence Length[J]. arXiv preprint arXiv:2105.15075, 2021.


**Summary Of The Paper:**

The paper proposes a token pooling approach based on the K-Means algorithm to improve the efficiency of vision transformers. The method is evaluated on image classification, where it reduces the computational complexity by half while maintaining similar performance.

**Summary Of The Review:**

The paper proposes a new token pooling method to achieve efficient vision transformers, which is interesting and well-motivated. But the method heavily relies on manual tuning and the generalization is limited. I will increase the rate if the concerns are well addressed.

---

> ### Author Response · Authors · 2021-11-15
> **Reply to the questions from Reviewer E53C from the authors**
>
> Thank you for your careful evaluation and valuable suggestions. We have answered each of your questions in the following.
>
> **[1] The K-Means clustering method for efficient transformers has been introduced in (Roy et al. 2021), which narrows the contribution of the paper.**
>
> This is a good question. We would like to clarify that the motivation and focus of our work and of (Roy et al., 2021) are very different.  Our contribution is a novel token downsampling algorithm that exploits redundancy in representations. The purpose of our method is to reduce the number of tokens. In contrast, Roy et al. focus on improving the $O(n^2)$ time complexity of softmax attention via sparsifying the attention Gram matrix with k-means and leave the number of tokens unchanged. As shown in Table 1, for vision transformers softmax attention is not the main bottleneck. Thus, utilizing the routing attention proposed by Roy et al. (2021) will not significantly improve the computational cost of vision transformers.
>
> For us, k-means and k-medoids are means to utilize redundancy among tokens, and they appear as solutions to minimize the reconstruction error in eq. (9).  Additionally, our analysis on softmax-attention being a high-dimensional Gaussian filtering can also serve as an explanation to the effectiveness of (Roy et al., 2021).  We revised the paper to explain the differences with (Roy et al., 2021) in Sec. 2.2.1.
>
>
>
> **[2] According to Table 2, 3 and 4, the proposed framework is sensitive to the selection of hyperparameters. It seems that the method is tricky and heavily relies on manual tuning.**
>
> We appreciate your concern however, quite the opposite, we don’t do any manual tuning of hyperparameters. Thank you for the question; we have revised the paper and clarified this in Appendix C.
>
> First, the proposed Token Pooling layer has only one parameter—the number of the output tokens $K$ (or equivalently the downsampling ratio). As such, it is the same as standard downsampling layers, such as max or average pooling, where the size of the kernel and strides play the same role as our parameter $K$.
>
> Second, our experiments use the method proposed by Goyal et al (2020) to automatically select $K$, i.e., we use a single parameter $\lambda$, which controls the overall efficiency of the model (please see Appendix C).  The process automatically determines $K$s for all layers. Please also notice that $\lambda$ is not tuned. We display results with all tested $\lambda$ to provide a family of models with different balances of flops/accuracy.
>
> All other hyperparameters are standard and inherited from DeiT.
>
>
> **[3] The method is only evaluated on the DeiT. Can it apply to other vision transformers, e.g., PVT and Swin?**
>
> Please see our reply in [General Reply 1](https://openreview.net/forum?id=EGtUVDm991w&noteId=gTyI53ucBvZ).  Since Token Pooling has no strong assumption on the input, it can be applied to transformers that use “global” attention mechanisms. We have revised the paper to clarify this in the abstract and contributions. For example, we can simply replace PVT’s patch embedding blocks with Token Pooling.  Nevertheless, it may require additional efforts to apply it on transformers that utilize “local” attention mechanisms like Swin.
>
>
> **[4] Reducing the number of queries is unfriendly to apply to downstream tasks.**
>
> In the paper we show improved flop/accuracy on ImageNet-1k and conduct extensive ablation studies. In general, it is believed that improving ImageNet results leads to improvements in downstream tasks. However, we agree that some downstream tasks like semantic segmentation or pose estimation may suffer from having fewer tokens if they rely on original grid structure. We have revised the paper to discuss this in Sec 2.2.2.
>
> We would also like to point out that reducing the number of queries/tokens may not necessarily sacrifice the performance or applicability of downstreamed tasks like semantic segmentation.  For example, Wu et al. (2021) used a small number of queries but still achieved state-of-the-art semantic segmentation results.
>
>
>
> **[5] It would be better to report the actual throughput and compare the proposed method with the previous dynamic vision transformers, such as (Rao et al. 2021) and (Wang et al. 2021).**
>
> Thank you for the suggestion and the related work [3]. We have added it to the related work. Please see our answer in General Reply 2 above.  We have calculated and included the throughput of our models in the revised paper (Appendix H).

---

### Official Review · Reviewer_RU8U · 2021-11-01

**Correctness:** 4
**Technical Novelty And Significance:** 4
**Empirical Novelty And Significance:** 3
**Recommendation:** 8
**Confidence:** 4

**Details Of Ethics Concerns:**

No ethical concerns to this paper.

**Main Review:**

Strengths:

+ Solid experimental results

Compared with a well-optimized DeiT baseline, token pooling significantly improves accuracy and reduces computation. Ablations in the appendix on different clustering, convolution downsampling, models are well written.

+ Well-structured analysis

The analysis in Table 1 clearly show the portion of different operators in vision transformer and figure 2 shows the limitation of score-based token downsampling, which motivates the authors to reduce redundancies with a novel token pooling method.

+ Impact on community

Vision transformers have been deployed on multiple vision tasks and are being a mainstream solution. Efficient vision transformers are important to be developed and the token pooling proposed in this paper can be used for other vision transformers without major modifications. It can replace convolution/patch-based downsampling and contribute to the community.

Weaknesses:

- Attention as a low-pass filter

I appreciate the analysis in section 3.3 on attention vs low-pass filter, but I don't get the relation between this observation and the token pooling methods. In the experiment at Appendix F, it seems token pooling is not affected by normalized or unnormalized Q/K vectors.

**Summary Of The Paper:**

In this paper, the authors propose a novel token-pooling method to reduce redundancies in tokens for recent vision transformers. They analyze the computation cost distribution of vision transformers and the limitations of grid-based & score-based token downsampling methods. They further formulate a reconstruction loss and optimize it with the token pooling layer. The experimental results based on DeiT show that token pooling improves accuracy while reducing computation cost by a large margin. This idea can be applied to other vision transformers as well.

**Summary Of The Review:**

Based on my main review above, I think this paper proposes a solid token pooling method for vision transformers that can be potentially applied to other vision transformers based on tokens. I recommend an acceptance to this paper.

---

> ### Author Response · Authors · 2021-11-15
> **Reply to the questions from Reviewer RU8U from the authors**
>
> Thank you for your careful evaluation and valuable suggestions.
>
> **“I appreciate the analysis in section 3.3 on attention as a low-pass filter...it seems token pooling is not affected by normalized or unnormalized Q/K vectors.“**
>
> Thank you for your appreciation. The purpose of the analysis is to show that the softmax attention acts as a low-pass filter therefore it’s output contains redundancies (i.e., tokens with similar feature values). Therefore token pooling can exploit these redundancies to reduce computation without sacrificing accuracy. For simplicity we provided this analysis for normalized Q/K vectors (even though in our experiments in the paper we used unnormalized Q/K vectors, which are used by standard transformers). As you have noted, in Appendix F, we further empirically show that Token Pooling works equally well for normalized or unnormalized Q/K vectors.

---

> > ### Comment · Reviewer_RU8U · 2021-11-18
> > **Comments on reply of the authors**
> >
> > Thanks for providing the reply based on our reviews.
> >
> > I look through all replies and they've addressed my concern on section 3.3. The authors further prove that token pooling can also be applied to local vision transformers like PVT and Swin.
> >
> > I think this token pooling method specialized for vision transformer is to interests of many researchers in the vision transformer community. I choose to keep my rating and recommend an acceptance to this paper.

---

### Official Review · Reviewer_jHZj · 2021-11-02

**Correctness:** 3
**Technical Novelty And Significance:** 3
**Empirical Novelty And Significance:** 2
**Recommendation:** 5
**Confidence:** 4

**Main Review:**

Strengths
1. The authors analyzed the computational cost of vision-transformer components in detail. These pie charts (in Table 1) visually illustrate the computational bottlenecks of vision transformers are fully-connected layers.
2. The authors demonstrated that softmax-attention is a low-pass filter under mild assumptions, thus attention layers will generate redundant representations.
3. This paper is well written and easy to understand. The authors provide a clear explanation and motivation behind their token pooling method. As far as I'm concerned, it has an impressive feature in that it adopts clustering algorithms (such as K-Means and K-Medoids) to reduce the number of tokens.

Weaknesses
1. Despite claims that the proposed token pooling is an effective operator that can benefit many architectures, but in this paper, the authors only examined based on the DeiT architecture. If token pooling is a general method for vision transformers, the paper should apply it to at least the current SOTA vision transformers.
2. As mentioned in the introduction, max pooling and average pooling are widely-used downsampling methods. From my experience, average pooling is a competitive method to reduce the number of tokens. Why not do an ablation study with max/average pooling?
3. Personally, I would appreciate experiments with more vision transformers. If the authors proved that the proposed token pooling could be applicable to other vision transformers besides DeiT, it would strengthen its value.
4. It would be better to provide more results and analysis of time consumption and inference speed, including the clustering process.



**Summary Of The Paper:**

This paper proposes a new token downsampling method for vision transformer, called Token Pooling, to prune redundant tokens efficiently, so as to achieve a better flop-accuracy trade-off. Specifically, token pooling is a nonuniform data-aware downsampling method, which uses cluster algorithms to aggregate information from tokens automatically. To keep important information, the authors also proposed minimizing the reconstruction loss caused by downsampling. The authors performed experiments on the ImageNet-1k dataset, showing that the proposed token pooling can significantly improve the flop-accuracy trade-off over the existing downsampling methods.

**Summary Of The Review:**

In summary, this work presents a new token downsampling method for efficient vision transformers. The paper is well written and properly structured, but the experiment is only performed on DeiT and ImageNet-1k benchmark, and no time analysis is provided, which is not up to par.

---

> ### Author Response · Authors · 2021-11-15
> **Reply to the questions from Reviewer jHZj from the authors**
>
> Thank you for your careful evaluation and valuable suggestions.  We have answered each of your questions in the following.
>
> **[1]  “The authors only examined the DeiT architecture. If token pooling is a general method for vision transformers, the paper should apply it to at least the current SOTA vision transformers.”**
>
> Please see our [General Reply 1](https://openreview.net/forum?id=EGtUVDm991w&noteId=gTyI53ucBvZ) above where we answer the question. Note, it was not exactly clear, which specific current SOTA transformers were referred to by the reviewer, so we've made our best effort to respond.
>
> In addition, our assertion of wide applicability on transformers using global attention stems from the fact that our method does not make strong assumptions about its input. We also rely on the property that the commonly used softmax attention is a low-pass filter that produces redundancy (i.e., tokens with similar feature values), as shown by our analysis. Motivated by the sampling theorem, we remove this redundancy by downsampling tokens while minimizing reconstruction loss (eq. 9).
> We would also like to point out that DeiT utilizes the standard transformer block, which is used in many state-of-the-art transformers. For example, our DeiT + Token Pooling is not very different from the architectures used in PoWER-BERT, PVT, DynamicViT, PiT, and HVT, when we replace their downsampling layers with ours.
>
> **[2] “an ablation study with max/average pooling”**
>
> In the paper, we show an ablation study on convolution with stride as one of the downsampling methods. It is a generalized version of average pooling. Average pooling can be implemented by convolving the feature map with a constant kernel of size n*n containing 1/n^2 and subsample (stride) every n pixels. By allowing the model to learn the kernels, we allow it to find a better kernel for the downsampling than average pooling. Therefore, we believe that our ablation study of “convolution downsampling” is sufficient for the community to compare typical pooling methods with our Token Pooling.  We have revised the paper to clarify this in Appendix D.
>
> **[3] “analysis of time consumption and inference speed was not reported”**
>
> Thank you for the suggestion. Please see our General Reply 2 above.  We have calculated and included the throughput of our models in the revised paper (Appendix H).  Please note that we listed the flops consumed by the clustering in Tables 2, 3, and 4 in the paper.

---

> > ### Comment · Reviewer_jHZj · 2021-11-22
> > **Reply to the authors**
> >
> > Thanks for your quick answer.
> >
> > > We can simply replace PVT’s (Wang et al., ICCV'21) patch embedding blocks with Token Pooling. Nevertheless, it may require additional effort to apply it on transformers that utilize “local” attention mechanisms like Swin transformers.
> > >
> > > Our DeiT + Token Pooling is not very different from the architectures used in PoWER-BERT, PVT, DynamicViT, PiT, and HVT, when we replace their downsampling layers with ours.
> >
> > Frankly, my concerns are mainly twofold:
> >
> > - In my opinion, DeiT is a relatively weak baseline (compared to other newer networks, e.g. Swin/PVTv2). I am more curious about whether token pooling is compatible with these more optimized models, so as to reduce their overhead without sacrificing too much accuracy. However, it seems that there are some prerequisites for using token pooling, e.g. global attention. Thus, this method seems difficult to apply to other models that use "local attention" or "local + global attention" (e.g., Swin/Twins).
> > - In addition, although the standard transformer block in DeiT is mainly-used, there is still a difference between these mentioned networks, i.e. the columnar (ViT/DeiT) or pyramid (PVTv2/PiT) structure. The author claims that applying token pooling to PVT is easy, but there are no experiment results. I would appreciate exploring whether token pooling still works well with pyramid-structured transformers. Because to some extent, the pyramid structure also reduces redundant patches.
> >
> > > Average pooling can be implemented by convolving the feature map with a constant kernel of size n*n containing 1/n^2 and subsample (stride) every n pixels. By allowing the model to learn the kernels, we allow it to find a better kernel for the downsampling than average pooling.
> >
> > I agree with this view.
> >
> > > Please see our General Reply 2 above. We have calculated and included the throughput of our models in the revised paper (Appendix H).
> >
> > Thanks for providing the throughput based on my reviews.

---

> > > ### Author Response · Authors · 2021-11-23
> > > **reply**
> > >
> > > Thank you for your comment.
> > >
> > > While it is not feasible to conduct and report additional experiments at this point, we would like to point out that even our DeiT-based implementation outperforms (or is on par with) several very recent and contemporeneus networks, such as pyramid architectures PVT, HVT, ConViT, SWIN, see Sec.1 in our General Reply and Fig.3b of the paper. In addition, we have compared with a pyramid downsampling strategy. Specifically, our “convolution” baseline is a form of a pyramid. We show that token pooling is better.
> > >
> > > We also would like to share a different perspective on the prerequisites. In our view, there is no prerequisite for using Token Pooling, rather there is a prerequisite for local attention mechanisms. More specifically, local attention mechanisms (e.g., Swin/Twins) make a stringent assumption about the input tokens structure, i.e. the arrangement into the image grid, which (we agree) may not be easily achieved with Token Pooling.
> > >
> > > Once again, thank you for your comments.

---

### Author Response · Authors · 2021-11-15
**General Reply by the authors**

We thank all the reviewers for their careful and valuable evaluation of our work and for pointing out the strengths of our work:
- Detailed analysis of computational cost of vision transformers (Reviewers jHZj, RU8U)
- Insights on softmax attention (Reviewers jHZj, E53C)
- Solid results and great potential impact on the community (Reviewer RU8U)
- Paper is well written and organized (Reviewers jHZj, E53C)


There are two common questions:

**1. Can Token Pooling be applied to architectures other than DeiT, for example, PVT or Swin?**

Token Pooling can be applied to transformers that use “global” attention mechanisms. We have revised the paper in the abstract and contribution. For example, we can simply replace PVT’s (Wang et al., ICCV'21) patch embedding blocks with Token Pooling.  Nevertheless, it may require additional effort to apply it on transformers that utilize “local” attention mechanisms like Swin transformers (Liu et al., ICCV'21).

We chose to use DeiT as our base architecture in the paper since it is a widely used baseline and its pipeline is well optimized and understood. Moreover, DeiT utilizes the standard transformer block, which is also commonly used and understood by the community.

Despite using the standard transformer block of DeiT, our results outperform PVT on ImageNet-1k (please see Table 5).  For example, PVT-Tiny is 75.5% at 1.9 GFlops, and ours is 79.4% at 1.9 GFlops; PVT-Small is 79.8% at 3.8 GFlops, and ours is 80.7% at 3 GFlops.  We also outperform various state-of-the-art methods (please see Figure 3) and are on par with Swin (Swin-T is 81.3% at 4.5 GFlops, ours is 81.4% at 4.4 GFlops).

We would also like to point out that in our ablation studies, we compare with the “convolution downsampling” method, which is used by PVT, Swin and other transformers like (Pan et al., 2021; Heo et al., 2021).  It is also a generalized version of the average pooling layer (which uses a constant convolution kernel).  Thus, our studies are informative and valuable, since they isolate the effect of the downsampling layers from the design of the transformer blocks. Our experiments clearly show that Token Pooling is a superior downsampling mechanism than the “convolution downsampling” method.


**2. What is the throughput (inference speed) of our model?**

We agree that throughput represents the actual speed during inference time, and it reflects the responsiveness of our models. Nevertheless, throughput also highly depends on the specific hardware, implementation quality, and the workload and the state (e.g. temperature) of the machines. Since it is usually difficult to control all these factors to have a fair comparison with other papers, in our original paper we chose to report the theoretical computational cost (i.e., flops) that is known to be highly correlated with the energy consumption of the models on device.

That being said, we agree with Reviewers jHZj and E53C that reporting the throughputs of our models, in addition, can serve as a reference for the community on how our models actually perform given our implementation and the type of GPU. Thus, we measured the throughputs of our models and put the results in the revised paper (please see Appendix H and Figure 12).  Note that our implementation is not optimized and thus has a large room for improvement.

We briefly summarize our findings:
With the same accuracy, using Token Pooling increases the throughput of DeiT-e318 by 25% (1640 vs 1310 images/sec), of DeiT-e252 by 22% (2190 vs 1791 images/sec), and of DeiT-Ti by 14% (3220 vs 2834 images/sec).
Our pytorch implementation of clustering algorithms can be improved, for example, via implementing a CUDA kernel.

As can be seen, despite our non-optimized code, Token Pooling significantly improves both the flops and throughputs of DeiT models.

We reply to individual questions from each reviewer in separate comments.

---

### Decision · Program_Chairs · 2022-01-20

**Decision:**

Reject

**Comment:**

This submission receives mixed ratings initially. Two reviewers lean negatively while one reviewer is positive. The raised issues include
whether the proposed method can be adapted to other vision transformers, the design choice of pooling strategy, the computational time cost, the similarity to an existing work, and the influence of the proposed method on downstream tasks. In the rebuttal, the authors have addressed several issues such as pooling strategy analysis and time consumption.

There are still some issues not completely solved. The proposed method introduces K-mean clustering on tokens between different layers. The K-mean clustering is prevalent and the weighted clustering does not make the technical contribution sufficient. Also as a general token pooling operation, the proposed method shall be integrated into various types of vision transformers (e.g., vanilla ViT [a], ConViT [b]), rather than one single DeiT. Besides, the downstream tasks in DeiT are not conducted in the proposed method.

Overall, the AC feels the proposed method, although interesting, requires a major revision that addresses existing issues.  The authors are suggested to further improve the current submission and welcome to submit for the next venue.

[a]. An Image is Worth 16x16 Words: Transformers for Image Recognition at Scale. Dosovitskiy et al. ICLR 2021.

[b]. ConViT: Improving Vision Transformers with Soft Convolutional Inductive Biases. Ascoli et al. ICML 2021.